# Comparative Analysis of Three Different Cooking Methods on Structures and Volatile Compounds of Fresh *Lyophyllum decastes*

**DOI:** 10.3390/foods14234106

**Published:** 2025-11-29

**Authors:** Xiaoshu Wu, Yan Wang, Weiyu Li, Chuannan Long, Jingjing Cui

**Affiliations:** 1School of Life Science, Jiangxi Science & Technology Normal University, Nanchang 330013, China; suzi010134@163.com (X.W.); yypz1003@163.com (Y.W.); 18379876567@163.com (W.L.); 2Jiangxi Key Laboratory of Bioprocess Engineering, Jiangxi Science & Technology Normal University, Nanchang 330013, China; 3Analysis and Testing Center, Jiangxi Science & Technology Normal University, Nanchang 330013, China; 4Jiangxi Key Laboratory of Organic Chemistry, Jiangxi Science and Technology Normal University, Nanchang 330013, China

**Keywords:** *Lyophyllum decastes*, cooking methods, volatile compounds, scanning electron microscopy (SEM), Fourier transform infrared spectroscopy (FTIR), comprehensive two-dimensional gas chromatography–mass spectrometry (GC×GC-MS), gas chromatography–ion mobility spectrometry (GC-IMS)

## Abstract

This study investigated the effects of three cooking methods—fragmenting process (FP), boiling treatment (BT), and high-pressure steam (HPS) treatment—on the structure and volatile compounds (VOCs) of fresh *Lyophyllum decastes*. The surface morphology and functional groups of *L. decastes* were analyzed by scanning electron microscopy (SEM) and Fourier transform infrared spectroscopy (FTIR), respectively. The VOCs in *L. decastes* were analyzed by comprehensive two-dimensional gas chromatography–mass spectrometry (GC×GC-MS) and gas chromatography–ion mobility spectrometry (GC-IMS). SEM results showed that HPS resulted in the most pronounced structural disruption, forming a honeycomb-like porous surface, whereas FP yielded smaller fragments with smoother surfaces. FTIR spectra indicated that none of the treatments significantly altered the characteristic functional groups. A total of 73 VOCs were identified by GC×GC-MS, including 23 hydrocarbons, 14 alcohols, 10 ketones, seven aldehydes, six ethers, three esters, two terpenes, and eight other compounds. Additionally, 22 VOCs were identified by GC-IMS, including seven alcohols, six aldehydes, five esters, three ketones, and one other compound. The four compounds benzaldehyde, benzeneacetaldehyde, (E)-2-hexen-1-ol, and 1-hexanal were detected by both methods. Among the three methods, FP induced the least structural damage and better preserved the VOCs. These results offer theoretical insights and technical support for the flavor-oriented deep processing of *L. decastes*.

## 1. Introduction

*Lyophyllum decastes*, a species within the Basidiomycetes class, Agaricales order, and Tricholomaceae family, is a rare edible mushroom prized for its high nutritional value and distinctive flavor. In its dried form, it contains 18.2% to 23.5% protein, including all eight essential amino acids. Notably, umami amino acids such as glutamic acid and aspartic acid constitute over 40% of the total amino acid content [1]. The polysaccharide content ranges from 6.8% to 10.3%, and these compounds exhibit multiple biological activities, including antioxidant, hypoglycemic, anti-inflammatory, and anti-cancer properties [2], thus exhibiting broad application prospects in functional foods and flavor-enhanced products.

In recent years, research on *L. decastes* has mainly focused on genome sequencing [3], polysaccharide extraction [2,4], and nutrient composition determination [1]. However, there is a paucity of research on the deep processing of *L. decastes*-derived products. Fresh *L. decastes* is highly perishable owing to its high moisture content (approximately 89% wet basis) [5], which readily induces quality deterioration. This necessitates the application of deep processing technologies to extend its shelf life while retaining its characteristic flavor, which is crucial for the development of the *L. decastes* product industry.

Numerous studies have confirmed that processing can regulate the composition and content of volatile compounds in edible mushrooms while also extending their shelf life and facilitating transportation and storage. For example, Yang et al. found that 57 volatile compounds (VOCs) could be identified in *L. decastes* treated with different drying methods, and the composition of flavor compounds changed significantly after drying [5]. Meanwhile, drying effectively resolves the perishability of *L. decastes* caused by high water content, facilitating its transportation. Additionally, this regulatory effect of processing on volatile compounds has been extensively verified in various other common edible mushrooms, such as *Agaricus bisporus* [6], *Lentinula edodes* [7], and *Lanmaoa asiatica* [8], further supporting the universality of processing-induced flavor modulation in edible mushrooms. However, research on fresh *L. decastes* processing has rarely been reported.

In the food processing industry, high-pressure steam (HPS), fragmenting process (FP), and boiling treatment (BT) are three widely used technologies. Their regulatory mechanisms on food quality are distinct: High-pressure steam loosens the cell-wall polysaccharide network through the synergistic effect of high temperature and pressure. For instance, a study on the stipe of *Flammulina velutipes* demonstrated that the steam explosion effect induced by high pressure can disrupt the dense cell structure, significantly increasing internal porosity and surface roughness, thereby promoting polysaccharide dissolution [9]. Additionally, it can alter component transformation pathways by regulating enzyme activity—high-pressure cooking can significantly enhance the antioxidant capacity of edible mushrooms [10], while in pork, lipoxygenase (LOX) can be completely inactivated at 50 °C and 600 MPa [11], a result that highlights the inhibitory effect of high pressure on biological enzymes. Fragmenting process (FP), on the other hand, destroys cell integrity, bringing intracellular enzymes, which were originally spatially separated, into full contact with their substrates. Grosshauser confirmed in a study on *Agaricus bisporus* that the enzyme systems within the tissue can efficiently promote the synthesis of characteristic flavor compounds such as 3-methylbutanal and phenylacetaldehyde after fresh-cut processing [12]. Boiling treatment (BT), however, tends to cause the evaporation of low-boiling-point VOCs; for example, the contents of alcohols in *Agaricus bisporus* and terpenoids in *Pleurotus ostreatus* both showed a significant downward trend after boiling [13]. If these three methods are adopted for the cooking of fresh *L. decastes*, it will not only simplify the processing procedure but also maximize the retention of its texture characteristics, which is of great practical significance for the industrial development of *L. decastes* deep-processed products.

The accurate characterization of VOCs in *L. decastes* products relies on efficient analytical techniques. Among these, comprehensive two-dimensional gas chromatography–mass spectrometry (GC×GC-MS) and gas chromatography–ion mobility spectrometry (GC-IMS) have emerged as important technical tools in the field of flavor research, leveraging their complementary advantages. GC×GC offers a significantly greater peak capacity than conventional GC, thereby enabling effective differentiation of structural isomers [14]. For instance, Tian et al. successfully identified 79 VOCs in the fermented grains of *Qingxiangxing Baijiu* using this technique [15]. Zhou et al. combined GC-IMS and GC×GC-TOF-MS to identify 183 and 245 VOCs, respectively, in *Morchella esculenta*, fully validating the complementarity of these two techniques in flavor analysis [16].

Until now, these techniques have not been jointly applied to the study of VOCs in processed products of *L. decastes*, nor has any research systematically elucidated the effects of the aforementioned cooking methods on its characteristic VOCs. Therefore, this study aims to integrate GC×GC-MS and GC-IMS to investigate the effects of three processing methods—fragmenting process (FP), boiling treatment (BT), and high-pressure steam (HPS) treatment—on VOCs in fresh *L. decastes*. Meanwhile, scanning electron microscopy (SEM) and Fourier transform infrared spectroscopy (FTIR) were used to characterize the potential effects of these cooking methods on the structure of *L. decastes.* Through this systematic research, this study intends to fill the existing research gap in the deep processing field of fresh *L. decastes* and provide theoretical support for the development of subsequent *L. decastes* deep-processed products (e.g., *L. decastes* biscuits, *L. decastes* steamed breads, *L. decastes* canned products, etc.).

## 2. Materials and Methods

### 2.1. Raw Material and Processing

All the fresh *L. decastes* used in this study were purchased from Jiangsu Gubentang Biotechnology Co., Ltd (Jiangsu Province, China). The fresh *L. decastes* were chilled at 4 °C after harvesting and transported under a temperature-controlled cold chain. Subsequently, 100 g of fresh *L. decastes* fruiting bodies was cut into 1 cm segments. The segments were then mixed with 500 g of ultrapure water and processed according to the procedure outlined in Table 1. Upon completion of the treatment, the samples were naturally cooled to room temperature.

### 2.2. Observation of Surface Morphology in L. decastes by SEM

The SEM procedure was conducted in accordance with the methods described by Nie [18] and Zhao et al. [19]. *L. decastes* samples treated with three different cooking methods were stored in a −80 °C freezer for 12 h, followed by freeze-drying using an LGJ-2A freeze-dryer (Beijing SihuanQihang Technology Co., Ltd., Beijing, China) under a cold-trap temperature of −40 °C and a vacuum pressure < 50 Pa. The dried samples were then cut into uniform slices measuring 5 mm × 5 mm × 10 mm and mounted onto the sample stage using conductive adhesive, with the outer surfaces of both the cap and stalk oriented upward. Subsequently, the samples were coated with a thin layer of gold for 30 s. Morphological observations were performed using an SU8600 scanning electron microscope (Hitachi High-Tech, Tokyo Metropolis, Japan) at an accelerating voltage of 5 kV under high-vacuum conditions.

### 2.3. Analysis of Functional Group Structure in L. decastes by FTIR

Freeze-dried samples of *L. decastes*—prepared using three different distinct cooking methods—were ground into a fine powder. The freeze-drying procedure adhered to the protocol outlined in Section 2.2. FTIR analysis was conducted in accordance with the methodology established by Ma et al. [20]. The freeze-dried samples were finely ground using a mortar and pestle. A suitable quantity of the resultant powder was then mixed with KBr, further ground, and compressed into pellets. The prepared pellets were analyzed with a Spotlight 200i Fourier transform infrared spectrometer (PerkinElmer, MA, USA) over a wavelength range of 4000–400 cm^−1^.

### 2.4. Analysis of Volatile Compounds in L. decastes by GC×GC-MS

Take 2.0 g of the sample and put it into a 20 mL headspace vial, then seal it with a lid. All samples were analyzed by GCMS-QP2020NX gas chromatography–mass spectrometry (Hitachi High-Tech, Japan), and the analysis method referred to that of Cheng et al. [21]. Maintain a constant temperature at 60 °C for 30 min. Insert the pretreated 2.2 cm extraction head (65 µm PDMS-DVB, Hitachi High-Tech, Tokyo, Japan) into the headspace vial for extraction for 20 min. Insert the optical fiber into the GC injection port at 250 °C through the splitless mode for desorption for 5 min. Determination conditions of GC×GC-MS: First-dimension chromatographic column is SH-Rtx-5sil MS (30 m × 0.25 mm × 0.25 µm, Hitachi High-Tech, Japan). Two-dimensional chromatographic column is BPX-50 (3.5 m × 0.10 mm × 0.10 µm, SGE, Victoria, Australia). The carrier gas is He (purity 99.999%), with a flow rate of 1.0 mL/min. The temperature program for the first one-dimensional column: The initial temperature is 40 °C, maintained for 5 min, then increased to 230 °C at a rate of 3 °C/min, and maintained for 5 min. Mass spectrometry conditions: electronic ionization (EI) source, ion source temperature 230 °C, mass spectrometry electron energy 70 eV, collection mass spectrometry scanning range 35–400 *m*/*z*. The identification of VOCs is determined by comparing mass spectrometry with standard compounds using the NIST mass spectrometry library (NIST 17).

### 2.5. Analysis of Volatile Compounds in L. decastes by GC-IMS

We took 2.0 g of the sample and placed it in a 20 mL headspace vial, then sealed it with a magnetic cap before analysis. We used HS-GC-IMS (flavor Spec^®^, Gesellschaft für Analytische Sensorsysteme mbH, Dortmund, Germany) based on 490 Micro Gas Chromatograph (Agilent, CA, USA), equipped with an automatic sampler (Solid Phase Micro Extraction, 57,330 U, Supelco, PA, USA) and headspace sampling device. Subsequently, the samples were incubated at 80 °C for 15 min at a rotational speed of 500 rpm/min. After incubation, 500 µL of headspace gas was automatically injected into the syringe at 65 °C through the injection needle (in non-fractionation mode). The sample was moved into the MXT-5 (15 m × 0.53 mm × 1 µm) metal capillary gas chromatography column by high-purity nitrogen (99.999%) as shown below: the initial flow rate was 2 mL/min, maintained for 2 min, and the flow rate was uniformly increased to 100 mL/min within 18 min and maintained for 30 min. Under a drift gas of 150 mL/min (nitrogen, 99.99% purity), the ions of analytes ionised were directed to the drift tube at 45 ◦C. The VOCs were identified by using Vocal software (version 0.4.03, G.A.S. Dortmund, Germany). The database of NIST14 DB-5/HP-5 in Germany was used to identify VOCs. The unknown is characterized by the RI value and drift time. C4–C9 ketone (Sinopharm Chemical Reagent Co., Ltd., Shanghai, China) is an external standard used for calculating the RI value of VOCs [22].

### 2.6. Statistical Analysis

All experiments were performed in triplicate, and data were expressed as mean ± standard deviation. Statistical analysis was performed using SPSS 27.0.1 (SPSS Inc., Chicago, IL, USA). Differences among the groups were considered significant at *p* < 0.05. The Venn diagram was created at https://www.omicstudio.cn/tool (accessed on 6 November 2025). The other results were visualized and analyzed using origin 2024 (Origin Lab, Northampton, MA, USA).

## 3. Results and Discussion

### 3.1. The Surface Morphology of Fresh L. decastes Cooked by Three Different Methods

In this study, we observed the microstructures of the caps and stalk of *L. decastes* cooked by three different methods (Figure 1). The results showed that different cooking methods had a significant impact on the microstructure of *L. decastes*. The surface of the CK cap is smooth with few wrinkles, the stalk tissue structure is dense, and the fibers are evenly distributed. FP completely destroyed the cap and stalk structure, forming a large number of irregular fragments with diameters ranging from 50 to 200 µm. This treatment exposed numerous fresh fracture surfaces, significantly increasing the specific surface area, and some pore structures could be observed on the fragments. BT caused wrinkles on the surface of the cap, blurred pore edges, and caused a reduction in diameter to 3–5 µm. The fibrous structure on the stalk surface contracted due to thermal denaturation, and the spacing decreased. HPS caused the cap surface to present a unique honeycomb-like porous structure with pore diameters ranging from 20 to 50 µm. The edges were irregular and interconnected, forming a network similar to a “sponge-like” structure, and caused cracks about 100 to 200 µm deep to appear on the stem surface.

These dramatic changes in the HPS group are likely attributable to the rapid expansion of steam under high temperature and pressure, generating substantial shear forces that irreversibly damage cell membrane integrity, break fibers, and create pores, thereby completely disrupting the dense native structure [23,24]. The observed structural alterations align with findings from studies on other vegetables, such as carrots, where steaming and boiling also led to cell dehydration and tissue separation, potentially due to capillary shrinkage stress caused by water evaporation during cooking [25].

### 3.2. FTIR Analysis of Fresh L. decastes Cooked by Three Different Methods

FTIR analysis (400–4000 cm^−1^) confirmed that the infrared spectra of fresh *L. decastes* treated by FP, BT, and HPS were consistent with CK (Figure 2). This indicates that none of the three treatments altered the characteristic functional groups of *L. decastes*. A broad and intense peak appeared at 3405 cm^−1^, corresponding to the stretching vibration of O-H bonds. This suggests strong intermolecular and intramolecular interactions between polysaccharide chains and phenolic compounds [26]. A narrow and weak peak near 2930 cm^−1^ was attributed to the stretching vibration of C-H bonds [26], mainly derived from lipids and polysaccharides. The sharp absorption peaks in the range of 1760–1600 cm^−1^ belonged to the stretching vibration of C=O in carbonyl compounds, which may be associated with the high content of ketone compounds [27] or proteins in the samples—consistent with the high protein content of *L. decastes* [1]. The peak around 1260 cm^−1^ was attributed to C–O stretching in acetyl groups or other related compounds. The significant absorption at 1150–1030 cm^−1^ corresponded to the stretching of C-O and C-C bonds or the bending of C-OH groups [23,28], mainly originating from cell-wall chitin and polysaccharides. The peaks in the range of 350–600 cm^−1^ corresponded to the skeletal ring vibration of pyranose [29]. These results indicate that *L. decastes* is mainly composed of polysaccharides, proteins, lipids, and chitin.

Further analysis of the peak intensity of the infrared spectral curves revealed that different treatment methods exerted varying degrees of influence on the chemical structure of *L. decastes*. The CK exhibited a distinct broad peak at 3400 cm^−1^, indicating a high content of hydroxyl groups and an intact structure. The characteristic peaks of polysaccharides in the 1150–1030 cm^−1^ region were clear, demonstrating that the polysaccharide structure remained undamaged. The obvious absorption peak of β-glycosidic bonds at 608 cm^−1^ suggested the integrity of the chitin structure. FP caused a significant reduction in the intensity of the infrared absorption peak at 605 cm^−1^. This might be attributed to the mechanical shear force breaking the β-glycosidic bonds, leading to the collapse of the pyranose ring skeleton structure. For the BT, the absorption intensity at 3400 cm^−1^ slightly decreased, which could be related to the dissolution of some soluble polysaccharides or proteins. In the HPS, the peak intensity at 3405 cm^−1^ significantly increased with a narrowed peak shape, and the content of free hydroxyl groups rose. This indicated that high temperature and high pressure might have caused the breakage of some hydroxyl groups or the loss of moisture. The decreased absorption peak intensity in the 1150–1030 cm^−1^ region might result from changes in the content and structural morphology of chitin under high temperature and high pressure [30]. The weakened peak at 608 cm^−1^ suggested the possible partial degradation of β-glycosidic bonds.

These changes in peak intensity indicated that FP had a minor impact on the structure, mainly causing physical damage. BT exerted a slight effect on the protein structure, while the polysaccharide structure remained relatively stable. HPS treatment significantly affected both protein and polysaccharide structures, potentially leading to the partial degradation of cell-wall chitin.

### 3.3. GC×GC-MS Analysis of Fresh L. decastes Cooked by Three Different Methods

In this study, GC×GC-MS was employed for the qualitative analysis of VOCs in *L. decastes* treated by three different cooking methods. As shown in Table 2, a total of 73 VOCs were identified, including 23 hydrocarbons, 14 alcohols, 10 ketones, seven aldehydes, six ethers, three esters, two terpenes, and eight other substances. Among them, the VOCs in CK were the most complex, with 57 compounds detected. A total of 45 and 37 VOCs were detected in FP and HPS, respectively. BT contained the fewest VOCs, with only 22 compounds detected.

The proportions of VOCs in *L. decastes* samples treated with different methods are shown in Figure 3. Among these four samples, aldehydes and alcohols had relatively high contents and are the major VOCs in *L. decastes*. These results are consistent with those of Yang et al. [5]. Aldehydes accounted for 32.78%, 24.94%, 30.77%, and 22.10% of the total VOCs in CK, FP, BT, and HPS, respectively. Alcohols accounted for 20.8%, 50.51%, 4.43%, and 18.74% of the total VOCs in CK, FP, BT, and HPS, respectively. The proportion of alcohols was relatively low in BT and slightly decreased in HPS as well. The possible reasons for these phenomena can be summarized as follows: (1) This may be because low-boiling-point alcohols volatilize significantly with water vapor during heating, resulting in a reduction in their residual amounts in the food [31]. (2) Alcohols degrade during boiling and are oxidized to aldehydes or ketones upon contact with oxygen. In contrast, HPS improves the heat transfer efficiency of steam through high pressure, enabling the interior of *L. decastes* to quickly reach the treatment temperature (121 °C) and avoiding the continuous oxidation of alcohols caused by long-term low-temperature preheating [32]. (3) The honeycomb structure formed in *L. decastes* by HPS can create “local retention spaces”, slowing down the volatilization rate of alcohols with steam, whereas the BT group exhibits dense tissue due to fiber shrinkage, making alcohols more likely to directly volatilize.

In FP, 1-octanol (4.32%), (E)-2-hexen-1-ol (34.99%), and nonane (7.22%) were detected at significantly higher levels compared with other groups, while no terpinolene was detected. These differences may be attributed to the promotion of biosynthetic pathways of alcohol-type VOCs in mushrooms by mechanical damage [33]. Terpinolene was also not detected in BT and HPS, which may be due to the thermal degradation of heat-sensitive aromatic compounds (such as terpenes) through oxidation and C-C bond cleavage during heat treatment [31]. The specific increases in pentadecane (6.95%), heptadecane (4.41%), and tetradecane (2.82%) in BT indicate that heating selectively promotes the formation of long-chain alkanes. The contents of heptadecane and tetradecane in HPS were also relatively high, but their overall concentrations were lower than those in BT, suggesting that HPS is less effective than BT in promoting the formation of these compounds. These findings indicate that fatty acids undergo thermal decomposition to produce alkanes under humid conditions [34].

In HPS, isophorone (2.42%), and 2-octenone (1.01%) were detected, implying that high pressure may alter the structure of key enzymes involved in aromatic compound synthesis, thereby increasing ketone production and affecting aroma formation [35]. Moreover, the increased levels of 3,4-hexanediol (14.16%), and benzaldehyde (8.23%) in HPS suggested that high temperature and pressure might disrupt cell-wall structures, activate enzyme systems, and promote the oxidative cleavage of unsaturated fatty acids [36], leading to increased aldehyde production. Notably, benzaldehyde was not detected in BT, indicating that it may be volatile or prone to decomposition under humid and hot conditions.

As can be seen from Table 2, among the 73 identified compounds, 18 have corresponding odor descriptions retrievable in the FEMA database. We plotted these 18 odor substances into a heatmap (Figure 4). Further analysis revealed that the contents of benzeneacetaldehyde, benzaldehyde, 1-undecanol, 2-ethyl-1-hexanol, dodecanal, 1-pentanol, linalool, 2-heptanone, 2-nonanone, terpinolene, and geranyl benzoate were relatively high in CK. These compounds mainly have bitter almond, pine, citrus, berry, and floral notes, which constitute the complex odor profile of the CK group. In FP, relatively high contents of 1-hexanol and 1-octanol were detected. These compounds mainly have banana, floral, bitter almond, and burnt matches aromas. In BT, nonanal had high contents, which endow boiled *L. decastes* with fat, floral, green, and lemon aromas. In HPS, the content of isophorone and 2-octanone were significantly high, which endows high-pressure steam-treated *L. decastes* with fat, fragrant, cedarwood, and spice aromas.

In summary, aldehydes and alcohols are the main VOCs in *L. decastes*. Among the three aforementioned treatment methods, the alcohol content in FP was significantly higher than that in BT and HPS. This may be because mechanical damage to *L. decastes* activates the enzyme system in cells, thereby promoting the significant synthesis of alcohols such as (E)-2-hexen-1-ol (34.99%). Benzeneacetaldehyde, benzaldehyde, 1-undecanol, 2-ethyl-1-hexanol, dodecanal, 1-pentanol, linalool, 2-heptanone, 2-nonanone, terpinolene, geranyl benzoate, 1-hexanol, and 1-octanol constitute the main flavor components of *L. decastes*. Different treatment methods have varying effects on its flavor substances: FP can increase the concentration of main flavor substances by promoting alcohol synthesis through mechanical damage; BT and HPS promote the thermal decomposition of fatty acids to generate hydrocarbons. This agreed with the FTIR results.

**Table 2 foods-14-04106-t002:** Qualitative results of GC×GC-MS data from fresh *L. decastes* cooked by three different methods.

Count	Type	Compound	Formula	CAS	Odor Description	Relative Percentage Content/%			
						CK	FP	BT	HPS
1	Hydrocarbons	3-Methyl-pentane	C_6_H_14_	96-14-0	ND	ND	0.14 ± 0.04 ^b^	ND	0.63 ± 0.11 ^a^
2		Butane	C_4_H_10_	106-97-8	ND	ND	ND	2.41 ± 1.60	ND
3		Propane	C_3_H_8_	74-98-6	ND	1.15 ± 0.13	ND	ND	ND
4		2-Bromo-2-methyl-butane	C_5_H_11_Br	507-36-8	ND	0.48 ± 0.16 ^a^	0.21 ± 0.03 ^b^	ND	ND
5		1-Octene	C_8_H_16_	111-66-0	ND	0.23 ± 0.07	ND	ND	ND
6		1,3,5,7-Cyclooctatetraene	C8H8	629-20-9	ND	0.39 ± 0.06	ND	ND	ND
7		1-Undecene	C_11_H_22_	821-95-4	ND	0.60 ± 0.56 ^a^	ND	ND	0.31 ± 0.17 ^a^
8		Nonane	C_9_H_20_	111-84-2	ND	ND	7.22 ± 0.82	ND	ND
9		Decane	C_10_H_22_	124-18-5	ND	ND	ND	ND	0.31 ± 0.05
10		Terpinolene	C_10_H_16_	586-62-9	Pine	0.51 ± 0.17	ND	ND	ND
11		1-Dodecene	C_12_H_24_	112-41-4	ND	1.04 ± 1.15 ^a^	0.24 ± 0.05 ^a^	ND	ND
12		Undecane	C_11_H_24_	1120-21-4	ND	1.24 ± 1.04 ^a^	ND	ND	1.33 ± 0.63 ^a^
13		1-Tetradecene	C_14_H_28_	1120-36-1	ND	0.36 ± 0.12 ^a^	0.18 ± 0.12 ^a^	ND	ND
14		1-Tridecene	C_13_H_26_	2437-56-1	ND	ND	0.15 ± 0.08	ND	ND
15		β-Methylstyrene	C_9_H_10_	637-50-3	ND	0.53 ± 0.15 ^b^	0.35 ± 0.08 ^b^	1.13 ± 0.16 ^a^	0.71 ± 0.28 ^b^
16		1,4-Diethyl-benzene	C_10_H_14_	105-05-5	ND	ND	0.19 ± 0.05 ^b^	ND	0.50 ± 0.05 ^a^
17		Azulene	C_10_H_8_	275-51-4	ND	0.42 ± 0.07 ^a^	0.22 ± 0.05 ^b^	ND	ND
18		Dodecane	C_12_H_26_	112-40-3	ND	0.28 ± 0.03 ^b^	ND	0.79 ± 0.31 ^a^	0.74 ± 0.21 ^a^
19		Tetradecane	C_14_H_30_	629-59-4	ND	ND	0.29 ± 0.12 ^b^	2.82 ± 1.10 ^a^	2.01 ± 1.00 ^ab^
20		Pentadecane	C1_5_H_32_	629-62-9	ND	2.13 ± 0.18 ^c^	1.55 ± 0.34 ^c^	6.95 ± 0.57 ^a^	4.36 ± 1.34 ^b^
21		Heptadecane	C_17_H_36_	629-78-7	ND	ND	0.76 ± 0.32 ^a^	4.41 ± 2.80 ^a^	2.61 ± 1.67 ^a^
22		Hexadecane	C_16_H_34_	544-76-3	ND	ND	ND	ND	2.67 ± 1.28
23		Heneicosane	C_21_H_44_	629-94-7	ND	ND	ND	0.62 ± 0.15	ND
24	Alcohols	Tert-butanol	C_4_H_10_O	75-65-0	ND	0.79 ± 0.15 ^c^	0.69 ± 0.26 ^c^	2.49 ± 0.41 ^a^	1.45 ± 0.19 ^b^
25		2-Methyl-3-pentanol	C_6_H_14_O	565-67-3	ND	0.36 ± 0.05	ND	ND	ND
26		Cyclopentanol	C_5_H_10_O	96-41-3	ND	0.55 ± 0.06 ^a^	0.46 ± 0.06 ^a^	ND	ND
27		1-Pentanol	C_5_H_12_O	71-41-0	Balsamic, Fruit, Green, Pungent, Yeast	0.55 ± 0.08 ^a^	0.20 ± 0.01 ^b^	ND	ND
28		1-Hexanol	C_6_H_14_O	111-27-3	Banana, Flower, Grass, Herb	2.93 ± 0.33 ^b^	5.67 ± 0.60 ^a^	ND	ND
29		3,4-Bis(4-hydroxyphenyl)-3,4-hexanediol	C_18_H_22_O_4_	7507-01-09	ND	4.54 ± 2.91 ^b^	2.00 ± 0.41 ^b^	ND	14.16 ± 0.55 ^a^
30		1-Octanol	C_8_H_18_O	111-87-5	Bitter Almond, Burnt Matches, Fat, Floral	1.29 ± 0.81 ^b^	4.32 ± 0.28 ^a^	ND	0.63 ± 0.19 ^b^
31		(Z)-2-Penten-1-ol	C_5_H_10_O	1576-95-0	ND	ND	ND	ND	0.75 ± 0.20
32		(E)-2-Hexen-1-ol	C_6_H_12_O	928-95-0	ND	4.53 ± 0.93 ^b^	34.99 ± 1.83 ^a^	ND	ND
33		9,10-Tetradecenol	C_14_H_28_O	52957-16-1	ND	2.50 ± 0.22	ND	ND	ND
34		1-(2-Butoxyethoxy)-2-Propanol	C_9_H_20_O_3_	124-16-3	ND	0.47 ± 0.10 ^bc^	0.32 ± 0.11 ^c^	1.17 ± 0.14 ^a^	0.84 ± 0.28 ^ab^
35		2-Ethyl-1-hexanol	C_8_H_18_O	104-76-7	Green, Rose	0.36 ± 0.04 ^a^	0.19 ± 0.01 ^b^	ND	ND
36		2-(Hydroxymethyl)-2-Nitro-1,3-propanediol	C_4_H_9_NO_5_	126-11-4	ND	0.22 ± 0.07 ^a^	0.12 ± 0.02 ^b^	ND	ND
37		1-Undecanol	C_11_H_24_O	112-42-5	Mandarin	1.72 ± 0.15 ^a^	1.55 ± 0.35 ^ab^	0.76 ± 0.09 ^c^	0.90 ± 0.47 ^bc^
38	Ketones	4-Methyl-2-hexanone	C_7_H_14_O	105-42-0	ND	0.55 ± 0.13	ND	ND	ND
39		Methyl isobutyl ketone	C_6_H_12_O	108-10-1	ND	0.38 ± 0.08	ND	ND	ND
40		2-Methyl-cyclopentanone	C_6_H_10_O	1120-72-5	ND	ND	0.34 ± 0.03	ND	ND
41		2-Heptanone	C_7_H_14_O	110-43-0	Blue Cheese, Fruit, Green, Nut, Spice	0.20 ± 0.03	ND	ND	ND
42		2-Dodecanone	C_12_H_24_O	6175-49-1	ND	0.22 ± 0.03	ND	ND	ND
43		2-Octanone	C_8_H_16_O	111-13-7	Fat, Fragrant, Mold	0.77 ± 0.14 ^a^	ND	ND	1.01 ± 0.21 ^a^
44		Isophorone	C_9_H_14_O	78-59-1	Cedarwood, Spice	ND	ND	ND	2.42 ± 1.55
45		4,4-Dimethyl-2-cyclohexen-1-one	C_8_H_12_O	1073-13-8	ND	0.42 ± 0.04	ND	ND	ND
46		2-Nonanone	C_9_H_18_O	821-55-6	Fragrant, Fruit, Green, Hot Milk	0.27 ± 0.03	ND	ND	ND
47		2-Undecanone	C_11_H_22_O	112-12-9	Fresh, Green, Orange, Rose	1.53 ± 0.10 ^a^	0.92 ± 0.17 ^b^	1.61 ± 0.30 ^a^	0.63 ± 0.16 ^b^
48	Aldehydes	1-hexanal	C_6_H_12_O	66-25-1	Apple, Fat, Fresh, Green, Oil	8.97 ± 0.48 ^a^	8.46 ± 0.58 ^a^	ND	1.92 ± 0.52 ^b^
49		2,3,6-Trichlorobenzaldehyde	C_7_H_3_C_l3_O	4659-47-6	ND	5.46 ± 0.73 ^c^	2.43 ± 0.33 ^c^	18.67 ± 2.51 ^a^	9.49 ± 0.84 ^b^
50		Nonanal	C_9_H_18_O	124-19-6	Fat, Floral, Green, Lemon	3.09 ± 1.36 ^b^	1.68 ± 0.56 ^b^	7.91 ± 1.38 ^a^	5.15 ± 3.39 ^ab^
51		Benzaldehyde	C_7_H_6_O	100-52-7	Bitter Almond, Burnt Sugar, Cherry, Malt, Roasted Pepper	9.81 ± 3.42 ^a^	8.23 ± 3.41 ^a^	ND	1.12 ± 0.29 ^b^
52		Undecanal	C_11_H_22_O	112-44-7	ND	1.32 ± 0.69 ^b^	3.82 ± 0.62 ^a^	2.46 ± 0.37 ^ab^	1.96 ± 0.69 ^b^
53		Benzeneacetaldehyde	C_8_H_8_O	122-78-1	Berry, Geranium, Honey, Nut, Pungent	1.78 ± 0.21 ^a^	0.33 ± 0.02 ^b^	1.73 ± 0.41 ^a^	1.34 ± 0.16 ^a^
54		Dodecanal	C_12_H_24_O	112-54-9	Citrus, Fat, Lily	2.34 ± 3.07 ^a^	ND	ND	1.12 ± 0.24 ^a^
55	Ethers	Methoxy-ethen	C_3_H_6_O	107-25-5	ND	ND	1.91 ± 1.16 ^c^	10.89 ± 1.42 ^a^	6.70 ± 1.81 ^b^
56		Allyl ethyl ether	C_5_H_10_O	557-31-3	ND	0.55 ± 0.32	ND	ND	ND
57		Dibutyl ether	C_8_H_18_O	142-96-1	ND	0.51 ± 0.06 ^ab^	0.34 ± 0.13 ^b^	ND	0.83 ± 0.25 ^a^
58		2-Heptyl-1,3-dioxolane	C_10_H_20_O_2_	4359-57-3	ND	1.34 ± 0.34	ND	ND	ND
59		N-Acetylcolchinol methyl ether	C_21_H_25_NO_5_	65967-01-3	ND	2.44 ± 1.40 ^bc^	0.71 ± 0.10 ^c^	4.06 ± 0.36 ^ab^	6.35 ± 1.86 ^a^
60		2-(2-Butoxyethoxy)-Ethanol	C_8_H_18_O_3_	112-34-5	ND	ND	0.21 ± 0.04	ND	ND
61	Esters	Sec-butyl nitrite	C_4_H_9_NO_2_	924-43-6	ND	0.78 ± 0.17 ^ab^	0.54 ± 0.24 ^b^	1.48 ± 0.32 ^a^	1.28 ± 0.56 ^ab^
62		Geranyl benzoate	C_17_H_22_O_2_	94-48-4	Floral	0.43 ± 0.19	ND	ND	ND
63		Propylene carbonate	C_4_H_6_O_3_	108-32-7	ND	0.39 ± 0.10 ^ab^	0.24 ± 0.06 ^b^	ND	0.44 ± 0.10 ^a^
64	Terpenoids	Retinol	C_20_H_28_O	116-31-4	ND	4.70 ± 0.92 ^a^	1.25 ± 0.08 ^b^	0.76 ± 0.17 ^b^	ND
65		Linalool	C_10_H_18_O	78-70-6	Coriander, Floral, Lavender, Lemon, Rose	1.60 ± 0.97 ^a^	0.17 ± 0.03 ^b^	ND	ND
66	Others	3-Pyrroline	C_4_H_7_N	109-96-6	ND	1.15 ± 0.06 ^a^	0.74 ± 0.18 ^b^	ND	ND
67		12,13-Dihydro-7H-dibenzo(a,g)carbazole	C_20_H_15_N	63077-00-9	ND	2.43 ± 0.75 ^a^	2.27 ± 0.84 ^a^	10.71 ± 7.57 ^a^	4.23 ± 0.74 ^a^
68		9,10-Dihydro-9,9-dimethyl- Acridine	C_15_H_14_N	53884-62-1	ND	0.57 ± 0.11 ^b^	0.32 ± 0.04 ^c^	ND	0.96 ± 0.13 ^a^
69		Dibenzyl ketoxime	C_14_H_13_NO	1788-31-4	ND	ND	0.23 ± 0.02	ND	ND
70		Vincamine	C_21_H_26_N_2_O_3_	1617-90-9	ND	6.63 ± 1.29 ^b^	ND	ND	8.98 ± 1.05 ^a^
71		Prednisolone acetate	C_23_H_30_O_6_	52-21-1	ND	1.89 ± 1.55	ND	ND	ND
72		Colchicine	C_22_H_25_NO_6_	64-86-8	ND	6.37 ± 1.41 ^b^	2.51 ± 0.48 ^b^	14.02 ± 4.25 ^a^	7.16 ± 1.11 ^b^
73		Beclomethasone	C_22_H_29_ClO_5_	4419-39-0	ND	0.96 ± 0.34 ^ab^	0.32 ± 0.03 ^b^	2.13 ± 0.81 ^a^	2.02 ± 1.00 ^a^

ND, not detected. The description of the compound’s odor is based on the relevant literature and research [37,38,39], as well as the FEMA database. Relative percentage content is expressed as mean and ±standard deviation of three determinations. Different superscript letters in the same row imply significant differences (*p* < 0.05).

### 3.4. GC-IMS Analysis of Fresh L. decastes Cooked by Three Different Methods

To thoroughly investigate the variation of VOCs under different processing methods, a qualitative analysis was conducted by comparing retention times and drift times with reference standards. Peak intensities are relative values from GC-IMS detection, reflecting the abundance of each compound; higher values indicate higher relative concentration. A total of 22 known VOCs were identified and categorized into five major classes: seven alcohols, six aldehydes, five esters, three ketones, and one other compound (Table 3). No specific compounds were identified for areas 1 to 17 in the database. Among the VOCs detected in *L. decastes*, alcohols accounted for the highest proportion (55.5%), followed by aldehydes (30.2%), which is consistent with the findings obtained via GC×GC-MS analysis.

To illustrate the differences in volatile profiles among the samples, characteristic fingerprints were established (Figure 5). Each row in the figure represents a sample or all signal peaks selected during processing. As shown in Figure 5, there were significant differences in the VOCs of the four different flavor samples. The peak signal intensity of VOCs in region A was significantly reduced after BT or HPS treatment. In contrast, the peak signal intensity in region B was significantly increased after FP treatment but remarkably decreased following BT or HPS treatment. Five compounds, namely, (E)-2-hexen-1-ol, (E)-2-heptenal, (E)-2-octenal, 1-butyl acetate, and 1-hexanal, were identified in region A. Among these, (E)-2-heptenal, (E)-2-octenal, 1-butyl acetate, and 1-hexanal contribute almond, fat, dandelion, and fruit odors to the CK and FP groups, respectively. In region B, six compounds—benzaldehyde, (Z)-3-hexenyl acetate, 3-heptanol, 2-acetyl-5-methylfuran, propanoic acid, 3-(methylthio)-, methyl ester, and furfuryl propionate—were identified, endowing the FP group with a complex and layered flavor profile featuring notes of bitter almond, herb, vegetable, nut, tropical fruit, floral, and spicy aromas. In addition, CK, FP, BT, and HPS samples also contained relatively high contents of 2-octanol, benzeneacetaldehyde, 2-methylpentanal, ethyl propanoate, and 3-hydroxy-2-butanone. These compounds impart fat, berry, apple, and butter aromas to the four types of samples.

Notably, the levels of (E)-2-hexen-1-ol and (Z)-3-hexenyl acetate in CK and FP were significantly higher than those in BT and HPS. This may be attributed to the activation of enzyme activity in cells and the initiation of cellular defense mechanisms when fresh *L. decastes* was damaged and exposed to oxygen, leading to a substantial increase in the synthesis of these two substances [33,44]. In FP, the peak signal intensity of benzaldehyde in *L. decastes* was significantly higher than BT and HPS, which is consistent with the previous results of FTIR and GC×GC-MS. Additionally, the signal intensity of 2-pentanone in the BT group was 1.4–24 times higher than in other groups, suggesting its potential as a characteristic marker compound for BT. This fruity aroma may originate from the synthesis of ketones under prolonged heating conditions [32]. The signal intensities of 2-butanone, 3-hydroxy-, and 2-hexanone in HPS significantly increased, indicating enhanced release under high-pressure conditions.

### 3.5. Comparative Analysis of Volatile Compounds by GC×GC-MS and GC-IMS

A comparative analysis was performed using GC×GC-MS and GC-IMS techniques, which identified 73 and 22 VOCs in the four groups of *L. decastes* samples, respectively. Among them, four VOCs (Figure 6), including benzaldehyde, benzeneacetaldehyde, (E)-2-hexen-1-ol, and 1-hexanal, were jointly identified by both detection methods. Additionally, both methods confirmed that alcohols and aldehydes are the main VOCs of *L. decastes*, and this result is consistent with the research findings of Yang et al. [5].

Both GC×GC-MS and GC-IMS have their own advantages in detecting VOCs. GC×GC-MS separates VOCs through two-dimensional column (SH-Rtx-5sil MS and BPX-50) chromatography and programmed temperature ramping (40–230 °C). Qualitative analysis is then performed by matching mass spectra to standard libraries (e.g., NIST) and interpreting fragment ions. In contrast, GC-IMS uses a single column, and the drift tube temperature of the ion mobility spectrometer (IMS) is fixed (45 °C). Qualitative analysis relies on detecting the ion drift time and retention time in an electric field, then matching to a custom-built IMS library.

Based on these characteristics, GC×GC-MS can detect more types of VOCs, especially exhibiting advantages in the identification of hydrocarbons, ethers, and terpenoids, and can more comprehensively reflect the overall composition of VOCs in *L. decastes*. It detects a wider variety of substances, many of which are different from those detected by GC-IMS. While GC-IMS has the advantages of fast detection speed (completion within 30 min per sample) and simple sample pretreatment, it complements the results of GC×GC-MS in the identification of ester compounds.

## 4. Conclusions

The three cooking methods (FP, BT, and HPS) did not alter the functional groups of *L. decastes*. FP broke the mushrooms into small fragments with smooth surfaces, BT caused surface shrinkage, and HPS promoted the formation of a unique honeycomb-like porous structure on the surface. GC×GC-MS identified 73 VOCs, while GC-IMS detected 22, among which 4 (benzaldehyde, benzeneacetaldehyde, (E)-2-hexen-1-ol, 1-hexanal) were identified by both methods. The results from these two analytical techniques indicated that alcohols and aldehydes are the main VOCs of *L. decastes*. Notably, the significant increase in (E)-2-hexen-1-ol and (Z)-3-hexenyl acetate in FP may be related to the defense mechanism after cell damage.

This study explored the flavor changes during the processing of *L. decastes*, providing practical technical parameters for the flavor design of deep-processed products of *L. decastes* and also offering theoretical and methodological references for optimizing the flavor characteristics of other edible mushrooms during processing.

## Figures and Tables

**Figure 1 foods-14-04106-f001:**
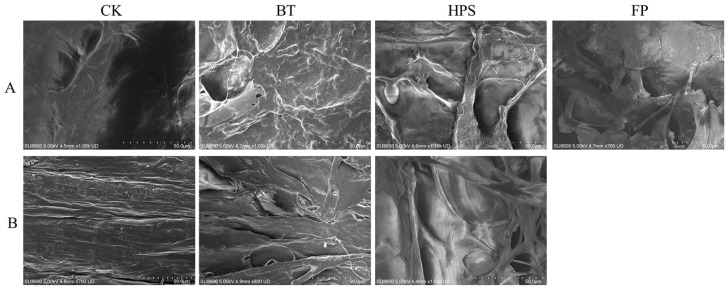
The effects of three different cooking methods on the surface morphology of fresh *L. decastes*. Cap of fruiting bodies (**A**). Stalk of fruiting bodies (**B**). The FP group was a mixture of cap and stalk fragmentation.

**Figure 2 foods-14-04106-f002:**
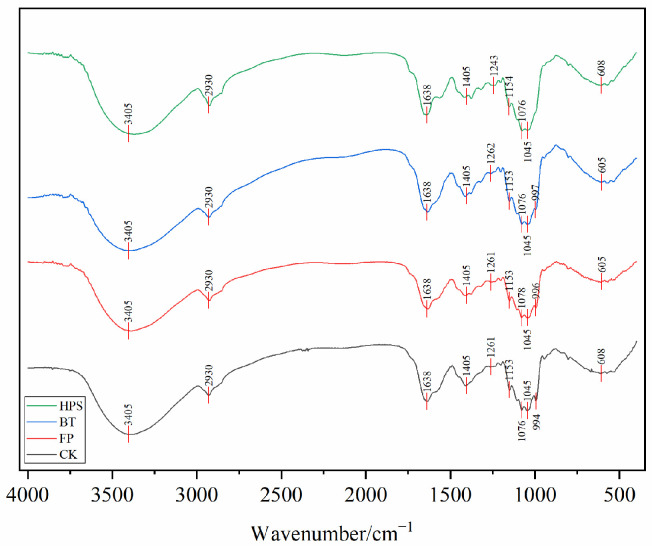
FTIR spectra of fresh *L. decastes* cooked three different ways.

**Figure 3 foods-14-04106-f003:**
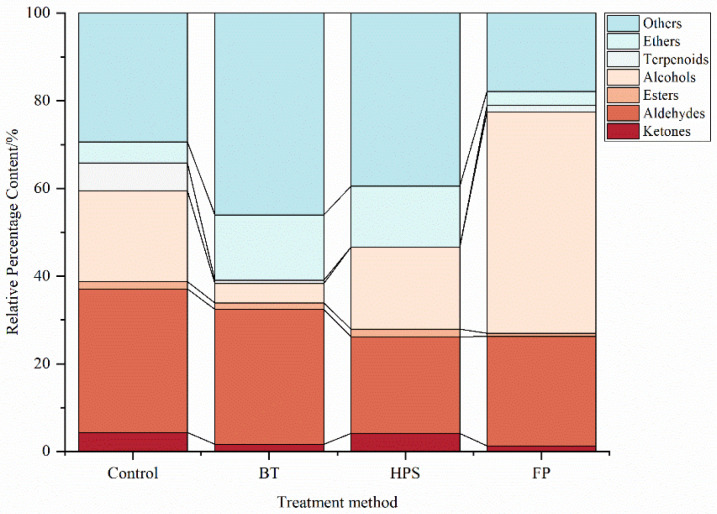
Percentage content of each substance in fresh *L. decastes* cooked by three different methods.

**Figure 4 foods-14-04106-f004:**
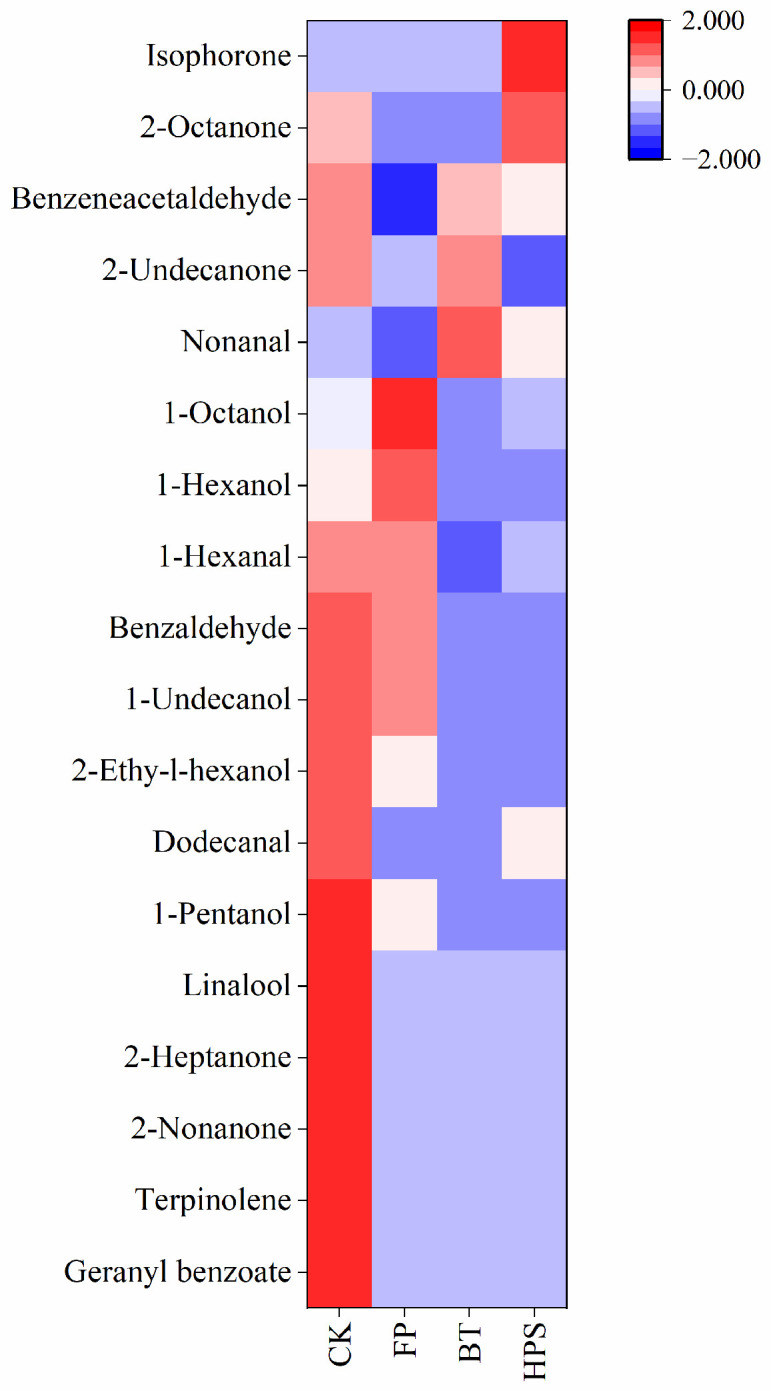
Heatmap analysis of volatile aroma compounds in fresh *L. decastes* cooked by three different methods.

**Figure 5 foods-14-04106-f005:**
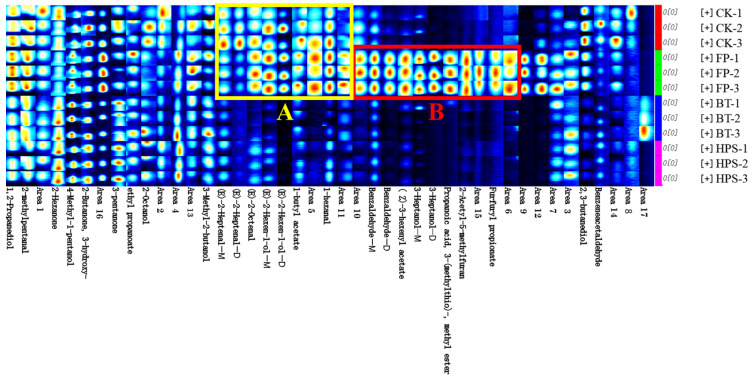
Fingerprint profiles of volatile compounds in fresh *L. decastes* cooked by three different methods.

**Figure 6 foods-14-04106-f006:**
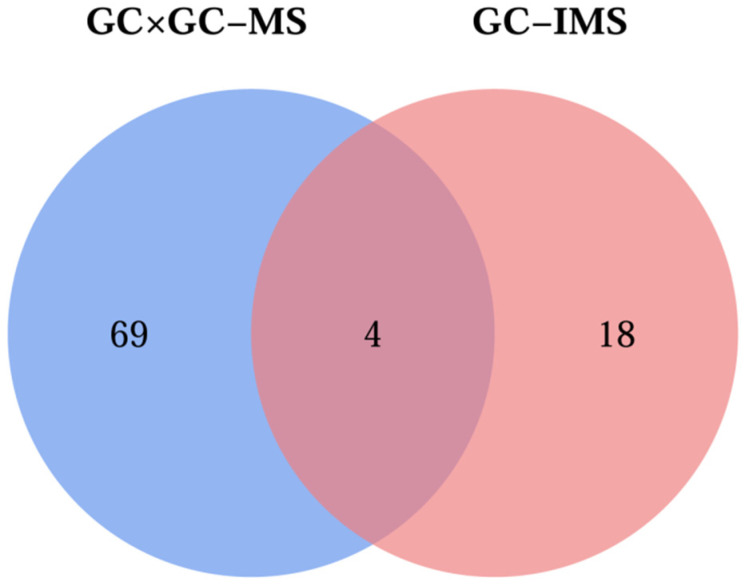
Venn diagram for GC×GC-MS and GC-IMS results of volatile compounds in fresh *L. decastes* cooked by three different methods.

**Table 1 foods-14-04106-t001:** Specific parameters for the processing of fresh *L. decastes* fruiting bodies.

Treatment Method	Specific Parameters
CK	No additional treatments were performed.
FP	The segments of fresh *L. decastes* were processed in a blender using the smoothie mode for 20 min.
BT	The segments of fresh *L. decastes* were boiled in a pot for 20 min after the water boiled [17].
HPS	The segments of fresh *L. decastes* were steamed in an autoclave at 121 °C for 20 min.

**Table 3 foods-14-04106-t003:** Volatile compounds identified by GC-IMS of fresh *L. decastes* cooked by three different ways.

Count	Type	Compound	CAS	Formula	RI	Odor Description	Peak Intensity			
							CK	FP	BT	HPS
1	Alcohols	3-Heptanol (M)	589-82-2	C_7_H_16_O	903.9	Herb	526.29 ± 69.33 ^b^	2548.27 ± 27.09 ^a^	354.14 ± 163.00 ^c^	260.15 ± 22.92 ^c^
2		3-Heptanol (D)	589-82-2	C_7_H_16_O	901.2	Herb	99.49 ± 10.80 ^b^	1187.37 ± 45.80 ^a^	88.53 ± 11.87 ^b^	84.42 ± 11.28 ^b^
3		(E)-2-Hexen-1-ol (M)	928-95-0	C_6_H_12_O	886.5	ND	717.71 ± 238.61 ^a^	927.15 ± 32.52 ^a^	136.62 ± 27.43 ^b^	218.39 ± 108.79 ^b^
4		(E)-2-Hexen-1-ol (D)	928-95-0	C_6_H_12_O	886.1	ND	731.62 ± 407.93 ^a^	1021.29 ± 25.57 ^a^	50.78 ± 2.27 ^b^	72.81 ± 20.05 ^b^
5		4-Methyl-1-pentanol	626-89-1	C_6_H_14_O	837.6	ND	1774.04 ± 299.45 ^b^	2590.30 ± 58.47 ^a^	1715.96 ± 94.00 ^b^	1736.45 ± 218.78 ^b^
6		3-Methyl-2-butanol	598-75-4	C_5_H_12_O	684.8	ND	345.86 ± 171.52 ^a^	255.40 ± 114.98 ^a^	317.51 ± 81.31 ^a^	344.41 ± 69.13 ^a^
7		2-Octanol	123-96-6	C_8_H_18_O	1019.7	Fat, Mushroom	256.12 ± 111.32 ^a^	145.67 ± 30.16 ^a^	203.09 ± 33.52 ^a^	261.72 ± 52.67 ^a^
8		1,2-Propanediol	57-55-6	C_3_H_8_O_2_	738.7	ND	576.00 ± 100.74 ^a^	670.68 ± 22.12 ^a^	543.78 ± 32.79 ^a^	269.60 ± 157.16 ^b^
9		2,3-butanediol	513-85-9	C_4_H_10_O_2_	740.2	ND	2252.22 ± 440.46 ^a^	1898.24 ± 74.40 ^a^	829.77 ± 117.01 ^b^	397.46 ± 232.33 ^b^
10	Aldehydes	Benzaldehyde (M)	100-52-7	C_7_H_6_O	983.5	Bitter Almond, Burnt Sugar, Cherry, Malt, Roasted Pepper	621.34 ± 118.51 ^b^	1458.17 ± 14.43 ^a^	483.27 ± 67.75 ^c^	569.53 ± 43.99 ^bc^
11		Benzaldehyde (D)	100-52-7	C_7_H_6_O	983.0	Bitter Almond, Burnt Sugar, Cherry, Malt, Roasted Pepper	135.97 ± 35.59 ^b^	720.75 ± 39.21 ^a^	75.92 ± 10.44 ^c^	78.36 ± 7.25 ^c^
12		(E)-2-Heptenal (M)	18829-55-5	C_7_H_12_O	977.6	Almond, Fat, Fruit	688.55 ± 131.25 ^a^	514.88 ± 20.17 ^b^	238.77 ± 24.94 ^c^	342.69 ± 16.10 ^c^
13		(E)-2-Heptenal (D)	18829-55-5	C_7_H_12_O	978.1	Almond, Fat, Fruit	474.58 ± 124.72 ^a^	331.94 ± 16.52 ^b^	84.63 ± 11.96 ^c^	119.24 ± 16.88 ^c^
14		(E)-2-Octenal	2548-87-0	C_8_H_14_O	1068.5	Dandelion, Fat, Fruit, Grass, Green, Spice	279.63 ± 33.88 ^a^	333.04 ± 14.61 ^a^	103.66 ± 15.95 ^b^	140.83 ± 59.73 ^b^
15		1-hexanal	66-25-1	C_6_H_12_O	813.6	Apple, Fat, Fresh, Green, Oil	535.93 ± 106.26 ^a^	635.36 ± 21.18 ^a^	134.95 ± 20.70 ^c^	274.72 ± 24.66 ^b^
16		Benzeneacetaldehyde	122-78-1	C_8_H_8_O	1053.6	Berry, Geranium, Honey, Nut, Pungent	567.57 ± 216.98 ^a^	370.03 ± 34.30 ^a^	444.22 ± 63.66 ^a^	424.81 ± 68.26 ^a^
17		2-methylpentanal	123-15-9	C_6_H_12_O	735.3	Green	529.13 ± 69.09 ^a^	591.00 ± 60.38 ^a^	580.89 ± 31.13 ^a^	146.12 ± 180.27 ^b^
18	Esters	(Z)-3-hexenyl acetate	3681-71-8	C_8_H_14_O_2_	1010.1	Banana, Candy, Floral, Green	191.80 ± 37.81 ^b^	925.34 ± 32.80 ^a^	141.78 ± 39.94 ^b^	198.43 ± 13.55 ^b^
19		Propanoic acid, 3-(methylthio)-, methyl ester	13532-18-8	C_5_H_10_O_2_S	1000.3	Cabbage, Garlic, Radish, Spice, Sulfur, Tropical Fruit	554.40 ± 80.59 ^b^	2228.40 ± 65.77 ^a^	875.65 ± 657.25 ^b^	414.71 ± 33.04 ^b^
20		1-butyl acetate	123-86-4	C_6_H_12_O_2_	800.2	Apple, Banana	368.86 ± 131.14 ^a^	448.41 ± 44.89 ^a^	231.31 ± 14.22 ^b^	214.15 ± 25.69 ^b^
21		ethyl propanoate	105-37-3	C_5_H_10_O_2_	677.8	Apple, Pineapple, Rum, Strawberry	688.44 ± 93.05 ^a^	720.01 ± 213.01 ^a^	559.26 ± 107.25 ^a^	565.81 ± 86.21 ^a^
22		Furfuryl propionate	623-19-8	C_8_H_10_O_3_	1087.3	Floral, Spice	84.64 ± 2.88 ^b^	281.86 ± 17.72 ^a^	79.70 ± 7.85 ^b^	77.87 ± 3.61 ^b^
23	Ketones	2-Hexanone	591-78-6	C_6_H_12_O	770.1	ND	969.86 ± 177.97 ^bc^	896.23 ± 37.21 ^c^	1220.71 ± 65.40 ^a^	1146.47 ± 48.14 ^ab^
24		2-Butanone, 3-hydroxy-	513-86-0	C_4_H_8_O_2_	713.2	Butter, Creamy, Green Pepper	1196.81 ± 263.13 ^ab^	971.37 ± 50.67 ^b^	1165.91 ± 65.37 ^ab^	1332.35 ± 60.76 ^a^
25		3-pentanone	96-22-0	C_5_H_10_O	678.6	ND	1632.96 ± 538.83 ^ab^	993.94 ± 393.07 ^b^	1416.93 ± 208.96 ^ab^	1695.13 ± 73.52 ^a^
26	Others	2-Acetyl-5-methylfuran	1193-79-9	C_7_H_8_O_2_	1015.5	Nut, Vegetable	167.94 ± 16.25 ^b^	1152.33 ± 14.29 ^a^	157.17 ± 29.99 ^b^	175.52 ± 17.06 ^b^
27		Area 1					266.21 ± 32.66 ^a^	258.84 ± 17.07 ^a^	210.27 ± 8.33 ^a^	221.38 ± 87.24 ^a^
28		Area 2					281.14 ± 32.43 ^a^	100.77 ± 20.52 ^b^	108.74 ± 14.39 ^b^	136.00 ± 33.00 ^b^
29		Area 3					29.20 ± 13.71 ^b^	73.52 ± 11.32 ^a^	23.56 ± 5.20 ^b^	34.64 ± 20.02 ^b^
30		Area 4					184.86 ± 56.20 ^ab^	231.58 ± 23.74 ^a^	112.93 ± 16.98 ^b^	207.49 ± 93.16 ^ab^
31		Area 5					35.30 ± 11.89 ^b^	49.05 ± 5.64 ^a^	6.33 ± 1.08 ^c^	10.22 ± 0.78 ^c^
32		Area 6					31.70 ± 2.79 ^b^	117.84 ± 36.08 ^a^	19.38 ± 1.70 ^b^	16.51 ± 1.14 ^b^
33		Area 7					64.11 ± 8.99 ^b^	150.80 ± 38.38 ^a^	81.06 ± 30.63 ^b^	78.33 ± 25.90 ^b^
34		Area 8					66.38 ± 34.58 ^a^	30.96 ± 6.51 ^b^	26.88 ± 2.73 ^b^	26.84 ± 1.58 ^b^
35		Area 9					262.06 ± 38.25 ^b^	3246.61 ± 101.38 ^a^	266.05 ± 47.08 ^b^	206.80 ± 11.74 ^b^
36		Area 10					122.90 ± 34.06 ^b^	227.66 ± 18.79 ^a^	39.90 ± 5.73 ^c^	56.85 ± 8.39 ^c^
37		Area 11					405.15 ± 108.75 ^b^	862.48 ± 86.14 ^a^	334.04 ± 21.14 ^bc^	240.96 ± 66.09 ^c^
38		Area 12					199.61 ± 91.01 ^b^	738.64 ± 155.30 ^a^	49.11 ± 6.65 ^b^	37.39 ± 6.53 ^b^
39		Area 13					330.29 ± 157.86 ^ab^	501.02 ± 90.29 ^a^	315.48 ± 38.59 ^ab^	161.73 ± 9.07 ^b^
40		Area 14					316.52 ± 79.50 ^a^	200.62 ± 37.50 ^b^	108.5 ± 8.02 ^c^	129.85 ± 3.72 ^bc^
41		Area 15					97.60 ± 14.87 ^b^	425.35 ± 66.74 ^a^	73.07 ± 10.99 ^b^	79.20 ± 11.43 ^b^
42		Area 16					2767.65 ± 581.89 ^b^	5132.04 ± 108.64 ^a^	1781.20 ± 201.44 ^c^	2154.29 ± 302.62 ^bc^
43		Area 17					81.68 ± 2.12 ^b^	59.52 ± 4.85 ^b^	282.41 ± 122.14 ^a^	83.34 ± 10.65 ^b^

ND, not detected. RI, retention index. The description of the compound’s odor is based on the relevant literature and research [40,41,42,43], as well as the FEMA database. Relative percentage content is expressed as mean and ±standard deviation of three determinations. Different superscript letters in the same row imply significant differences (*p* < 0.05).

## Data Availability

The original contributions presented in the study are included in the article; further inquiries can be directed to the corresponding authors.

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
