# Peer review of "Comparative Analysis of Three Different Cooking Methods on Structures and Volatile Compounds of Fresh Lyophyllum decastes"

_foods, 2025, doi:10.3390/foods14234106_

Round 1
Reviewer 1 Report
Comments and Suggestions for Authors
The study "Comparative Analysis of Three Different Cooking Methods on Structures and Volatile Compounds of Fresh Lyophyllum decastes" evaluated how fragmenting, boiling, and high-pressure steam treatments affect the structure and volatile compounds of L. decastes. High-pressure steam caused the most damage, while fragmenting best preserved structure and aroma. The results provide guidance for flavor-oriented processing of L. decastes.
General comment: Please check the Latin names — for some reason, the genus and species names are merged.
The introduction is somewhat unclear and lacks a coherent flow. It should be carefully revised to focus on the main topic of the manuscript. Some parts, particularly those related to GC-MS, could be moved to the Discussion section. The Introduction should also highlight the importance, significance, and novelty of the study.
Material and method: Please provide more information about the plant material used in the research. I suggest merging subsections 2.1 and 2.2 into one, titled “Raw Material and Processing.”
The Results and Discussion section is well written. Please ensure that the names of VOCs in the text are written in lowercase rather than capitalized, as they are now (Benzaldehyde, Benzeneacetaldehyde, (E)-2-Hexen-1-ol, 1-hexanal...).
Reviewer 2 Report
Comments and Suggestions for Authors
The manuscript titled “Comparative Analysis of Three Different Cooking Methods on Structures and Volatile Compounds of Fresh Lyophyllum decastes” investigates the effects of three processing methods on the structure and volatile compounds of fresh Lyophyllum decastes.
The study offers useful comparative data on L. decastes but does not sufficiently highlight its novelty relative to prior studies on other edible mushrooms (e.g., Agaricus bisporus, Lentinula edodes). The authors should emphasize what makes L. decastes unique in terms of its chemical profile or industrial potential.
The introduction is poorly written and lacks clarity. It does not provide a coherent background or adequately justify the purpose of the study. Key concepts are introduced in a disorganized manner, and important references or context that would help the reader understand the significance of the research are missing. Additionally, the logical flow between sentences and paragraphs is weak, making it difficult to follow the main argument. The introduction would benefit from restructuring, clearer definitions of terms, and a more concise explanation of the research gap and objectives.
The manuscript contains numerous grammatical errors, typographical mistakes, and inconsistencies throughout. Even the scientific name of the fungi is written incorrectly in several places (e.g., “Lyophyllumdecastes” instead of the correct Lyophyllum decastes). This error appears in the title, abstract, and keywords and should be corrected. In many places, words are merged together with no spaces between them. The authors are strongly advised to have the entire manuscript professionally edited by a native English-speaking scientific editor.
The Materials and Methods section is presented in an instructional, “cookbook-like” format (e.g., “Take 100 g of fresh L. decastes fruiting bodies and cut them into 1 cm small sections…”). This style is inconsistent with standard scientific writing, which requires passive, impersonal, and descriptive phrasing. The procedures should be rewritten as objective statements that describe what was done, not as instructions.
Line 212: In the presented FTIR spectra, the Y axis lacks a defined unit, and the stacked plotting prevents a mutual vertical scale. While this layout helps visualize spectral shape, it precludes quantitative comparison of transmittance between samples.
Lines 203-206: There are two statements in the explanation of the FTIR spectrum that cannot be stated with certainty. The first is “The sharp absorption peaks in the range of 1760–1600 cm⁻¹ belonged to the stretching vibration of C=O in carbonyl compounds, which might be related to the high content of ketone compounds in the samples [27].” While this range is characteristic of carbonyl groups, it cannot be attributed solely to ketones. For example, the band at 1638 cm⁻¹ corresponds to the C=O stretching vibration of amide I in proteins, which are among the most abundant compounds in this material. Additionally, the relevance of reference [27] to this statement is unclear. The second statement is: “The peak near 1260 cm⁻¹ was associated with the C=O ester bond of acetyl groups.” This band may instead correspond to C–O stretching and could arise from several classes of compounds, not exclusively from acetyl esters.
Line 231: The statement The absence of the peak in the range of 994–997 cm⁻¹ requires further investigation’ is inaccurate. The band does not disappear; it is still present but appears as a shoulder at higher intensity.”
The retention indexes in table 2 are unusually low, which suggests a mistake. Typically, they range from 600 to 3500. Also, experimental values need to be compared to literature ones to confirm the structure. Furthermore, the presence of compounds such as colchicine and beclomethasone in the GC-MS table suggests potential misidentification.
Authors need to explain the large difference in the results obtained by the two methods. GC-IMS detected 22 compounds, whereas GCxGC-MS detected 73 compounds.
Detected compounds in table 3 from 27 to 43: what are they? The relative percentages in the table 3 are very high, higher than 100%?
The manuscript has significant deficiencies that preclude publication. The study lacks clear novelty, the introduction and methods are poorly written, and the data contain major technical issues, including misinterpreted FTIR spectra, implausible retention indices, and inconsistent GC-MS/GC-IMS compound identifications, pervasive grammatical errors. Overall, the manuscript does not meet the standards required for publication and should be rejected.
Comments on the Quality of English Language
The manuscript contains numerous grammatical errors, typographical mistakes, and inconsistencies throughout. Even the scientific name of the fungi is written incorrectly in several places (e.g., “Lyophyllumdecastes” instead of the correct Lyophyllum decastes). This error appears in the title, abstract, and keywords, and should be corrected. In many places, words are merged together with no spaces between them. The authors are strongly advised to have the entire manuscript professionally edited by a native English-speaking scientific editor.
The Materials and Methods section is presented in an instructional, “cookbook-like” format (e.g., “Take 100 g of fresh L. decastes fruiting bodies and cut them into 1 cm small sections…”). This style is inconsistent with standard scientific writing, which requires passive, impersonal, and descriptive phrasing. The procedures should be rewritten as objective statements that describe what was done, not as instructions.
Reviewer 3 Report
Comments and Suggestions for Authors
This article discusses a comparative analysis of three different cooking methods on the structure and volatile compounds of fresh Lyophyllum decastes. This article is of interest to readers interested in the use of Lyophyllum decastes in processing and product development. However, some improvements are needed to ensure completeness of the content.
- Abstract: The abstract of a research article should be concise and comprehensive in covering all aspects of the research. Its primary purpose is to help readers quickly understand the overall nature of the research. It should include key findings or data from the research.
- Keywords should be checked. Keywords help readers and search engines identify and find articles quickly and efficiently. Keywords should clearly identify the main content and key points of the research.
- Introduction: Information on the chemical composition and active compounds found in Lyophyllum decastes should be included.
- Introduction: Research related to the effects of cooking methods on mushroom quality should be included.
- Introduction: The information should be complete. The introduction of the research article should state the research problem, objectives, and overview of the research being presented. It should include information on the key strengths or findings of this research.
- 2.1. Materials: Information regarding the source, shelf life, and post-storage handling of Lyophyllum decastes should be described.
- Table 1. Specific parameters for the processing of fresh L. decastes fruiting bodies: should be changed to Figure for clear explanation.
- It is important to check that the writing of units is consistent throughout the article, such as units of time and volume.
- Results and Discussion: “Further analysis of the peak intensity of the infrared spectral curves revealed that different treatment methods exerted varying degrees of influence on the chemical structure of L. decastes,” Academic information should be added to explain the details and related research.
- Results and Discussion: “Among these four samples, aldehydes and alcohols had relatively high contents and are the major volatile compounds in L. decastes,” Academic information should be added to explain the details and related research.
- All Table and Figure: It is recommended to standardize the decimal places for all numerical values in the figures and tables (to either the second or third decimal place).
- Figure 5. Fingerprint profiles of volatile components in fresh L. decastes cooked by three different methods: The font size should be adjusted to be clearer.
- Conclusion: Should additional explanations be provided regarding the consistency of the evidence and arguments presented and the relevance of the main questions raised?
- Conclusion: Details of the benefits to readers of this study and how future research can be developed should be included.
- There are quite a lot of abbreviations. Should be added an abbreviation section.
- Should be checked the formatting of reference documents.
Round 2
Reviewer 2 Report
Comments and Suggestions for Authors
The authors have responded thoughtfully to the reviewers’ suggestions, and the manuscript is now improved and suitable for publication.
Reviewer 3 Report
Comments and Suggestions for Authors
The content has been edited based on feedback.
This article is suitable for publication in the Foods journal.